# What’s in a Name? Chronic Vestibular Migraine or Persistent Postural Perceptual Dizziness?

**DOI:** 10.3390/brainsci13121692

**Published:** 2023-12-07

**Authors:** Alexander A. Tarnutzer, Diego Kaski

**Affiliations:** 1Neurology, Cantonal Hospital of Baden, 5404 Baden, Switzerland; 2Faculty of Medicine, University of Zurich, 8006 Zurich, Switzerland; 3SENSE Research Unit, Department of Clinical and Movement Neurosciences, Institute of Neurology, University College London, 33 Queen Square, London WC1N 3BG, UK; d.kaski@ucl.ac.uk

**Keywords:** chronic vestibular syndrome, dizziness, headache, vertigo, somatoform, migraine, mal de débarquement

## Abstract

Current consensus diagnostic criteria for vestibular migraine (VM) describes this as an episodic disorder. However, a minority of patients report prolonged (>72 h duration) or even persistent VM symptoms, prompting whether a chronic variant of vestibular migraine (CVM) should be introduced to the current classification and how best to define it. Here we summarize current evidence of such a potential chronic variant of VM and critically review proposed definitions for CVM. Potential approaches to establish a diagnostic framework for CVM include (a) following the distinction between episodic and chronic migraine headaches, namely, frequent and/or prolonged episodes of VM (but not persistent vertigo or dizziness) in the context of chronic migraine headaches or (b) daily dizzy spells over more than 6 months that responded well to prophylactic anti-migraine therapy. A key challenge when defining diagnostic criteria for CVM is how to distinguish it from other chronic vestibular syndromes such as motion sickness, persistent postural-perceptual dizziness (PPPD), and mal de débarquement syndrome. Indeed, more than 50% of patients with PPPD and up to 46% with mal de débarquement syndrome fulfil diagnostic criteria for episodic migraine headaches, suggesting these disorders may all lie along a spectrum. We propose that when VM becomes persistent, it is best classified as PPPD but that VM and PPPD are not mutually exclusive, such that patients with PPPD need not have features of VM, and the triggering event for persistent dizziness may be independent of migraine. However, further research is needed to better characterize the spectrum of clinical phenotypes in patients with chronic dizziness, migraine headaches and anxiety, to define whether a chronic variant of VM sufficiently differs from current persistent dizziness definitions.

## 1. Background

Episodic migraine (with or without aura symptoms) has a high prevalence worldwide, affecting 11.7% of the entire population (women: 17.1%, men 5.6%) [1]. Between 12 and 52% of patients diagnosed with episodic migraine report dizzy spells associated with migraine [2,3,4]. The relationship between migraine and vertigo is now well established [5,6], with diagnostic criteria for vestibular migraine (VM) proposed by the classification committee of the Bárány Society in 2012 [7], revised in 2021 [8] and also included in the appendix (A 1.6.6.) of the third edition of the International Classification of Headache Disorders (ICHD) [9]. Importantly, VM was conceptualized as an episodic disorder, with recurrent dizzy spells lasting between 5 min and 72 h and accompanying migrainous features in at least 50% of the vestibular episodes (Table 1). The reported lifetime prevalence of VM is between 1 and 2.7% [6,10], and it has been estimated that about 10% of all patients newly diagnosed with migraine headaches also meet diagnostic criteria for VM [11].

Overall, the clinical spectrum of VM is broad, with patients describing a range of dizziness symptoms, accompanying symptoms and duration of attacks [12,13]. A subset of patients with VM have prolonged attacks lasting weeks or even longer, exceeding the defined upper limit of 72 h [14,15]. Thus, with current diagnostic criteria, these patients cannot be diagnosed with VM according to ICHD 3 A.1.6.6 [9]. This raises the question of whether the diagnostic criteria for VM requires further revision, or whether a chronic variant of VM should be introduced, as for chronic migraine headaches (ICHD 3 1.3) [9]. However, when dizziness symptoms become more prolonged, they merge with other chronic vestibular syndromes such as persistent postural-perceptual dizziness (PPPD, [16]) and mal de débarquement syndrome (MdDS, [17]).

In this critical review, we summarize the current literature on VM with particular focus on patients presenting with prolonged VM episodes and discuss whether this subset of patients represents a separate entity or rather lies within a spectrum of episodic-to-chronic vestibular syndromes (PPPD).

### Review of the Literature

We have critically reviewed the published literature on Medline (accessed by Pubmed on 23 October 2023), using search terms including “vestibular migraine”, “chronic”, “persistent postural-perceptual dizziness”, “PPPD”, “migraine”, “mal de débarquement syndrome”, “vertigo” and “dizziness”. We focused on publications reporting original data on the spectrum of clinical presentations in VM, differentiation from chronic vestibular syndromes (including PPPD and MdDS) and new diagnostic criteria for VM variants. Manuscripts were identified by a full-text read, and relevant information was extracted by one author (AAT). Manuscripts included were screened for further citations suitable for this review. We have also screened published consensus papers and reviews on vestibular migraine (and related terms), PPPD and MdDS for further relevant citations.

## 2. Prolonged VM Episodes and Overlap with Chronic Dizziness

We have identified several publications reporting on the duration and/or frequency of VM episodes. However, we found only two studies that included patients classified as suffering from a chronic variant of VM [18,19]. Waterston proposed diagnostic criteria for “chronic migrainous vertigo” and reported on a cohort of 16 patients that had chronic dizziness for at least six months, that was thought to be migrainous in nature and that demonstrated strong treatment response to migraine prophylactic treatment [18]. A total of 6 out of these 16 patients had a history of migraine headaches, whereas actual headaches did not meet diagnostic criteria for migraine headaches in most cases. In 8 out of 16 cases, the dizzy spells were triggered by self-motion (7/16) and/or visual motion (4/16), and 11/13 cases (missing information in 3 cases) had a positive history of motion sickness. Importantly, this study is limited by a treatment–response selection bias and by the fact that in 62% of cases, it was not possible to make a diagnosis of present or past migraine according to the criteria of the International Headache Society (mainly because of lack of sufficient detail reported in the clinical presentation of headache episodes).

Chae and colleagues recently reported on a cohort of dizzy patients with a history of migraine headaches prospectively recruited in a tertiary dizzy clinic. They focused on the spectrum of clinical vestibular presentations and diagnoses (such as PPPD, MdDS, benign paroxysmal positional vertigo [BPPV] or Menière’s disease [MD]) but did not state whether they recruited consecutive patients; thus, they were potentially open to enrichment of chronic VM patients. These authors provided a framework for distinguishing between an episodic and a chronic variant of VM, thus allowing for a better characterization of VM subtypes and evaluating a more inclusive diagnostic VM scheme [19]. In brief, the proposed new subtype (definitive chronic VM) requires a more frequent occurrence (>15 days per month) of both vestibular symptoms and migraine episodes, or only for vestibular symptoms (>15 days per month) for the probable chronic VM variant. From the 54 patients included in this cohort, 33 were classified as having definitive (22/54, 41%) or probable (11/54, 20%) chronic VM, whereas the remaining 21 patients received a diagnosis of episodic VM (definitive = 10/54 [19%]; probable = 11/54 [20%]). Most patients in the chronic VM group reported vestibular episodes lasting more than 72 h (definitive CVM = 50%; probable CVM = 73%), and 79% of all definitive CVM patients and 60% of all probable CVM patients indicated constant (defined as being present “most or all the time”) vestibular symptoms. Furthermore, small fractions of all CVM patients also reported vestibular episodes lasting less than 5 min (definitive CVM = 9%; probable CVM = 18%). Overall, 59% of definitive CVM patients and 91% of probable CVM patients did not meet the *Bárány* Society diagnostic criteria for VM based on symptom duration [8]. Assessing the patients’ quality of life related to their vestibular symptoms, the CVM patients had significantly higher scores in the Vestibular Migraine Patient Assessment Tool and Handicap Inventory (VM-PATHI), reflecting higher disease severity compared to the episodic VM group. From the 22 patients with definite CVM, 4 patients also met diagnostic criteria for PPPD (*n* = 2, 9%) or MdDS (*n* = 2, 9%). Notably, diagnostic criteria for PPPD were not met in the vast majority of patients studied by Chae and colleagues because of an absence of worsening of symptoms with upright posture, suggesting there may be subtle but distinct pathophysiological mechanisms across CVM and PPPD or indeed that this feature of PPPD may be too restrictive.

Several other studies provided more detail on the spectrum of duration of dizzy spells in patients with established VM according to current definitions. In a cohort of 61 patients with definitive VM (according to the criteria proposed by [5]), Radtke and colleagues found that 52% of patients suffered from dizzy spells lasting more than 24 h. Of note, a subset of seven patients with sensorineural hearing loss also had a prolonged attack with a duration of 3 days or more (with a maximal episode duration of 42 days) and demonstrated some features compatible with MD [20]. The distinction between MD and VM in such cases is very challenging, and the decision to consider these patients as VM cases rather than MD cases by the authors was mainly driven by the pattern of hearing loss observed (being mild and more pronounced at higher frequencies in VM). The value of such atypical presentations remains debatable. Likewise, in a more recent study focusing on the clinical features of VM patients, Young and colleagues reported a duration of vertigo symptoms ranging from seconds to weeks, including some patients with persistent symptoms [14]. Specifically, in about 7% of VM patients in this study, the longest dizzy episode lasted weeks and another 8% indicated persistent vertigo.

Beh and colleagues described frequent interictal vestibular symptoms in a cohort of 131 patients meeting diagnostic criteria for VM [12]. A vast majority reported vision-induced dizziness (VID; 88.6%) or head-motion dizziness (HMD; 65.6%). About half of all VM patients (67/131, 51.1%) included in this study indicated persistent, almost constant dizziness between VM attacks. While the majority of these patients received a diagnosis of PPPD (64.2%) and MdDS (13.4%), the authors also considered the possibility of a chronic VM variant [12]. Beh and colleagues considered such a chronic VM variant the vestibular equivalent of a “chronic migraine” with at least 15 days of vestibular symptoms (instead of frequent headaches) per month—with superimposed attacks of vestibular symptoms accompanied by migrainous symptoms—for more than three months [12]. Likewise, in a cohort of VM patients, Eggers and colleagues identified a fraction of 22/52 (42%) patients with definitive VM that had comorbid neurological conditions (such as chronic subjective dizziness or tension-type headaches) and reported continuous dizziness or unsteadiness [21].

In a cohort of patients with chronic dizziness (defined as persistent non-spinning vertigo or dizziness for more than six months accompanied by hypersensitivity to motion stimuli and exacerbated by position change) [22], 23/73 (32%) patients also received a diagnosis of VM (based on the criteria of Neuhauser and Lempert [23]). Thus, this subset of patients with chronic dizziness presented with recurrent exacerbations of their vertigo or dizziness in conjunction with migraine headaches that closely resemble PPPD with superimposed VM rather than a single distinct entity of chronic dizziness.

## 3. Proposed Diagnostic Criteria for a Chronic Variant of VM

In contrast to existing criteria for VM, which consider this disease as an episodic syndrome, some authors have proposed new diagnostic criteria for a chronic variant of VM. These proposed diagnostic criteria conceptually follow different lines of argument.

For a diagnosis of “chronic migrainous vertigo”, Waterston required the presence of chronic dizziness for at least six months with either daily symptoms or bouts of weeks to months duration with only short symptom-free periods, a history of headaches (not necessarily meeting diagnostic criteria for migraine headaches according to the International Headache Society (IHS)) and a (near) total (i.e., >90% subjective improvement) response to prophylactic migraine therapy [18]. Patients with significant auditory involvement were excluded. A case series of 16 patients meeting these diagnostic criteria was presented by Waterstone in the same publication (for details see previous section). Here, however, there is a reliance on treatment response, but it is well recognized that many patients with VM do not respond to (nor may require) pharmacological therapies. As such, this diagnostic proposal ignores the holistic approach to VM management that is almost always required [24].

In contrast, Chae and colleagues proposed to broaden the diagnostic criteria for VM and to distinguish between an episodic VM variant and a chronic VM variant based on the frequency of occurrence of both vestibular episodes and migraine headaches [19]. This definition of chronic VM conceptually aligns with the requirements for chronic migraine headaches (ICHD 3 1.3 [9]) by requiring a minimal number of headache days (≥15 days) and days with vestibular symptoms (>15 days) per month for more than three months. As for current VM diagnostic criteria [8], Chae and colleagues differentiated between definitive and probable chronic VM. This distinction was based on (i) whether all four criteria for VM (see Table 1, A–D) were met (or only three, i.e., either B or C) and (ii) whether diagnostic criteria for a chronic migraine (ICHD 3 1.3) were achieved or if only a history of migraine headaches or chronic vestibular symptoms with migrainous features were present. Patients falling short of these two requirements were classified as having a probable chronic VM. The current definition of VM [8] does not, however, consider the frequency of vestibular symptoms as a diagnostic criterium except that at least five episodes must have occurred at some point in the past.

In the headache literature, status migrainosus is defined by the International Headache Society (ICHD 3 1.4.1 [9]) as a single, prolonged migraine episode lasting more than 72 h. A vestibular variant of this has not yet been proposed, but this term perhaps allows the construction of a bridge between a short episode of dizziness and established chronic dizziness symptoms (Figure 1).

### Treatment Strategies for the Subset of Patients with a Presumed Chronic Variant of VM

There is a lack of high-quality, randomized, controlled, prospective and double-blinded treatment trials in VM [24], limiting current treatment recommendations. In those studies that have reported on a chronic variant of VM, either response to treatment was not reported [19] or response to treatment was an inclusion criterion [18].

## 4. Chronic Vestibular Syndromes including PPPD and MdDS

The differential diagnosis for chronic dizziness is broad and includes bilateral vestibulopathy, drug-induced dizziness, neurogenetic disorders (e.g., cerebellar ataxia with neuropathy and vestibular areflexia syndrome [CANVAS], spinocerebellar ataxia), neurodegenerative disorders such as multiple systems atrophy (cerebellar variant), and functional dizziness syndromes such as PPPD and MdDS. In many patients suffering from functional chronic dizziness, an anecdotal link to migraine headaches and VM has been identified [25]. Thus, these disorders may share pathophysiological aspects and may represent different entities aligned on a spectrum of migraine-associated disorders. Thus, we review functional chronic vestibular disorders in more detail, focusing on their link to migraine headaches and VM.

### 4.1. Persistent Postural-Perceptual Dizziness (PPPD)

Diagnostic criteria for PPPD [16] characterize this as a chronic vestibular disorder that manifests with waxing and waning symptoms of dizziness, unsteadiness, or non-spinning vertigo causing significant distress or functional impairment (Table 2). Minimal symptom duration is three months with symptoms being present on most days. An exacerbation of symptoms in an upright posture, when exposed to passive or active motion and when exposed to moving visual stimuli and complex visual patterns is mandatory. There is a broad range of triggers for PPPD, i.e., conditions that result in vestibular symptoms or disrupt or threaten balance function, including neuro-otologic (e.g., acute unilateral vestibulopathy, BPPV, VM) and other medical conditions and psychological distress (including panic attacks and generalized anxiety). The reported incidence of PPPD 3–12 months after such an acute or episodic vestibular trigger (despite compensation or recovery from the initial disease) is about 25% [26]. Pathophysiological mechanisms leading to PPPD are not fully understood, but current concepts suggest abnormal bottom-up central processing of self and external motion signals that influence top-down postural behavior in predisposed individuals (see [27] for a detailed review of current evidence). Abnormal sway patterns on posturography were noted in 50% of PPPD patients in one study [28], whilst others have identified postural misperception as a possible clinical biomarker that better distinguishes PPPD from other chronic vestibulopathies [29].

In a retrospective survey study including 36 consecutive patients with PPPD, 19 patients (53%) also met diagnostic criteria for migraine headaches, and 6 of those 19 patients (17% of all patients) met diagnostic criteria for definitive VM [25]. An additional 31% of PPPD patients studied fulfilled three or four out of five criteria for migraine headaches, thus underlining an association between PPPD and migraine. In other studies, 26% [30] and 17% [31] of PPPD patients also met diagnostic criteria for migraine headaches, where VM seemed to act as a trigger and/or perpetuating factor for PPPD [25,28].

Indeed, VM seems to be a frequent trigger for PPPD. In a retrospective database analysis, VM was the second most frequent somatic trigger for PPPD (24%) after BPPV (27%) [28]. Likewise, in a retrospective review of 198 PPPD cases, the most common vestibular precipitating condition was VM, being present in 25% of patients [32]. When following-up patients diagnosed with BPPV for about 3.5 months, the risk for developing PPPD was significantly higher in those patients that also suffered from VM, whereas no such increase was observed in patients with migraine headaches (but no VM) [33].

Treatment of PPPD is often multimodal, combining vestibular rehabilitation therapy, cognitive behavioral therapy and antidepressants (SSRIs or SNRIs). While there have been no high-quality, randomized controlled trials published so far, more than 1000 PPPD patients have been included in treatment studies since 2002 [27]. Unlike treatment for VM where prophylactic agents are used to reduce the frequency and severity of attacks, therapeutic strategies for PPPD (and perhaps, by extension, for chronic migraine variants) work to reduce bodily hypervigilance and normalize mis-processing of self and visual motion signals through gradual re-exposure, thus emphasizing the value of cognitive physical therapy approaches early on [26] or even their preferential use over medications (see [34] and [Popkirov et al. in press]).

### 4.2. Mal de Débarquement Syndrome

The pivotal symptom of mal de débarquement syndrome (MdDS) is a persistent subjective sensation of self-motion (typically a rocking sensation like being on a boat). While transient mal de débarquement is common and usually resolves within 48 h (and never persists for more than one month), MdDS exceeds one month duration [35]. Recently, diagnostic criteria have been proposed by the classification committee of the Bárány Society [17] (Table 3). MdDS may emerge (a) after exposure to passive motion (sea travel, air travel, land travel) or (b) spontaneously. When re-exposed to passive motion, patients typically report transient improvement. This transient amelioration is the defining feature of MdDS and allows a distinction between MdDS and PPPD, with the latter leading to almost constant dizziness that becomes worse during passive motion [16], although most patients with PPPD will also report being symptom free when driving, making this distinction less clear. A link between migraine headache and VM and MdDS has been reported. In a retrospective study including 62 MdDS patients (87% women; spontaneous-onset MdDS: 55%), 39 patients (63%) met diagnostic criteria for VM as well [35]. While air travel and sea travel were the most frequent triggers for motion-triggered MdDS (22/28), a VM attack was the most commonly reported trigger for spontaneous-onset MdDS (8/22). Other symptoms reported in this cohort of MdDS patients (*n* = 62) included visually induced dizziness (68%) and head motion-induced dizziness (73%). Notably, clinical characteristics of the subgroup of MdDS with VM were substantially different from the characteristics of the subgroup without VM. Specifically, in the MdDS group with VM, age at symptom onset was lower, interictal visually-induced dizziness and head motion-induced dizziness were more frequent, and the dizziness handicap inventory (DHI) score was higher. Optimal disease control was achieved with monotherapy in 80% of patients, with venlafaxine (27%) and antiepileptics (11%) most frequently used, although in this publication, which specific medication(s) were given was not reported [35].

One area of contention is whether PPPD and MdDS share a common neuropathological mechanism, and thus, whether there is redundancy in using two separate terms, accepting that PPPD and MdDS may have different triggers, and some differing clinical characteristics (greater symptoms when standing still (MdDS) and on self-movement (PPPD)). We would argue, however, that the treatment approach for both conditions is in fact very similar, reflecting these share underlying mechanisms, with subtle individualized changes required for any given patient.

## 5. Discussion

The transition from an episodic to a chronic vestibular syndrome is often insidious. An acute isolated or recurrent episodic vestibular disorder may lead to residual, persistent vestibular symptoms even with recovery of the underlying acute event(s). Thus, patients may harbour a single chronic vestibular disorder (e.g., PPPD) in isolation, or with an ongoing other vestibulopathy (e.g., VM). Diagnosis of the persistent symptoms may, therefore, be challenging for the clinician because there may be more than a single explanation for the clinical picture, and even more challenging for the patient to understand. A central theme that requires exploration is whether labelling persistent vestibular symptoms as chronic VM or PPPD is a semantic deliberation with little clinical impact, or whether there are indeed fundamental clinical differences (greater anxiety, higher postural threat, etc.) requiring different treatment protocols. Ultimately, however, it is the underlying mechanism of the symptoms that will dictate the most appropriate therapeutic approach to maximize chances of a positive outcome. Treating a patient with persistent migrainous symptoms with preventative medications (for VM), is less likely to be effective than a more holistic approach that targets the persistent perceptual and psychological variables, whatever the label chosen. The important interplay between migraine, anxiety and dizziness has been elegantly described in the literature [36] and such features may help guide individualized therapies.

With a focus on the role of migraine headaches and especially VM within the spectrum of vestibular symptoms (Figure 1), we have critically reviewed the literature on the range of clinical presentations of and interplay between VM and chronic functional vestibular disorders such as PPPD and MdDS. The duration of vestibular symptoms in migraine patients varies substantially. We identified subgroups of patients meeting diagnostic criteria for VM reporting sometimes prolonged or even persistent episodes (e.g., [14]) and subgroups of migraine patients presenting with dizzy spells too short or—more often—too long to meet diagnostic criteria for VM [19]. At the same time, studies reporting on the characteristics of patients meeting diagnostic criteria for PPPD or MdDS indicated high rates of patients that also met the criteria for VM. In the following section, we will critically discuss how this overlap affects the diagnostic classification of patients with prolonged or persistent vestibular symptoms and provide suggestions to achieve consensus.

Patients suffering from episodic migraine (with or without aura) may notice an increase in headache frequency, and thus, may then meet diagnostic criteria for chronic migraine (with or without aura) according to the International Headache Society. Such a “chronification” of episodic migraine has been reported at a frequency of about 2.5–3% per year [37] and may also be seen for vestibular symptoms. Conceptually, there are several possible mechanisms to account for how patients may develop chronic dizziness, as discussed by Bronstein and colleagues [38]. Whereas episodic vertigo (e.g., due to BPPV, MD or VM) may be masked by continuous symptoms, recovery from an acute-onset vertigo may be incomplete (e.g., after acute unilateral vestibulopathy) or slowly progressive (or continuous and unchanging) symptoms without acute onset (e.g., in bilateral vestibulopathy) may emerge. Abnormal sensory processing, lowered perceptual thresholds for motion perception and increased visual motion sensitivity have been demonstrated in patients with VM [39]. This preponderance may facilitate the development of persistent symptoms, which shares similar alterations in the processing of multisensory input and internal weighting. Abnormal processing of repeated external stimuli (e.g., visual information) in the inter-ictal phase is a recognized pathophysiological feature of migraine, known as habituation deficiency [40]. Such features may also play in role in the chronification of dizziness symptoms in VM.

Obviously, preventing such a transition from a more limited, episodic vestibular syndrome to a chronic vestibular syndrome would have a substantial impact on the patient’s quality of life. Thus, identifying risk factors for such a chronification will be important, including premorbid anxiety, neuroticism, negative illness behaviors etc. Conceptually, patients that are “transitioning” from an episodic vestibular syndrome to a chronic vestibular syndrome such as PPPD or MdDS should be identified early (e.g., at the stage of an MdDS in evolution [17] or before the three-month minimal symptom duration for PPPD is reached) to prevent chronification. Such patients may be considered to be “at risk” of PPPD, offering a window to prevent this for becoming estabished.

With regards to the diagnostic criteria proposed for a chronic variant of VM, they are not optimized to recognize patients at risk. Rather they will detect patients that have already advanced to a (sub)chronic stage, with a symptom duration of at least 3–6 months. Consideration of the concept of “in transition to a chronic vestibular syndrome” as for those established for MdDS, could help shift the focus to prevention of chronification. Furthermore, the patient populations that will be included by applying either the definition proposed by Waterston [18] or by Chae and colleagues [19] will be substantially different. Whereas patients meeting the criteria proposed by Waterston will resemble those that suffer from other chronic vestibular syndromes such as PPPD and MdDS, those included when applying the criteria proposed by Chae and colleagues rather suffer from a frequent episodic vestibular syndrome.

There might be pathophysiological similarities between spontaneous-onset MdDS and PPPD, as both conditions are typically triggered by anxiety, vestibular events, or physical illnesses and share similar treatments [35]. A key question is how frequently patients meeting diagnostic criteria for definitive chronic VM also meet criteria for PPPD, motion sickness [41] or MdDS. As emphasized by Chae and colleagues, 31/33 chronic VM patients from their study did not meet diagnostic criteria for PPPD as no worsening of vestibular symptoms was reported with an upright position [19]. This would suggest that the patient population included in the CVM group in this study are distinct from PPPD patients and thus would require separate diagnostic criteria.

For an identification of patients at risk for chronification of vestibular symptoms, the definition brought forward by Chae and colleagues might be of value. Conceptually, those patients that have already transitioned from an infrequent episodic vestibular syndrome to a more frequent and longer duration of attacks, (though still episodic) may be at increased risk of developing persistent vertigo or dizziness (as required for MdDS and PPPD). On the other hand, patients meeting the diagnostic criteria proposed by Waterston more closely resemble patients suffering from PPPD or MdDS. Thus, these criteria will be less helpful in identifying patients at risk for chronification.

Moreover, current treatment approaches for VM [24], PPPD [26] and MdDS [42] are overlapping in relation to pharmacotherapy, especially when considering antidepressant medications. Thus, treatment response to preventative migraine medications does not necessarily prove a diagnosis of VM, as assumed by Waterston [18].

Establishing diagnostic criteria for chronic VM and considering it as a disease entity on its own has been discussed controversially in the literature. While Chae and colleagues advocated for more inclusive diagnostic criteria for VM to improve diagnosis and treatment [19], others have questioned the clinical value of providing diagnostic criteria for a separate, chronic variant of VM. Beh, for example, concluded that such separate diagnostic criteria may only serve to confound clinical trial data analyses [43].

Taking together the clinical data available, emphasizing the substantial overlap between chronic functional vestibular disorders and migraine headaches/VM, we would challenge the approach to add separate diagnostic criteria for a chronic variant of VM. Nonetheless, we do agree that patients with migraine headaches and prolonged vestibular symptoms not meeting current diagnostic criteria for VM need to be classified based on consensus diagnostic criteria. One potential strategy would be to revise current PPPD criteria, critically reviewing whether the required worsening of symptoms when standing up is a strict defining feature of PPPD or whether it could be considered optional. Alternatively, future clinical data may identify distinct pathophysiological mechanisms in patients with VM who develop persistent dizziness, versus those that develop PPPD in the absence of migraine features.

## 6. Limitations

As this was a critical review, we did not perform a systematic review of the literature. Therefore, we cannot exclude that we may have missed relevant publications. Recognizing this risk, we screened published review articles on vestibular migraine (and its predecessor terms), PPPD and MdDS. Importantly, the existing literature on the transition from episodic vestibular syndromes to chronic vestibular syndromes is limited and existing diagnostic criteria (for VM and PPPD) may have neglected certain subgroups including those with prolonged/persistent vestibular symptoms not fitting the current PPPD criteria. Thus, future prospective studies are needed to gain more knowledge about the clinical presentation and the disease course of such subgroups.

For patients, distinguishing between conditions with episodic vestibular symptoms (and symptom-free intervals) and chronic vestibular symptoms with varying intensity may be challenging. Thus, structured history addressing specifically the duration of single episodes with vestibular symptoms, their frequency and the presence/absence of symptom-free intervals is important to decide whether the diagnostic criteria for VM or PPPD are met. This will also be critical for future studies further delineating the clinical spectrum of patients with prolonged vestibular symptoms and VM.

## 7. Future Directions

With an established diagnosis of PPPD, these patients are already in a chronic stage; thus earlier diagnosis and potential early treatment have been missed. Therefore, future studies should especially target patient populations with acute/subacute vestibular symptoms that are still evolving and have not yet become chronic. In this patient group, treatment approaches are expected to have a better effect as central maladaptive mechanisms and sensory reweighting have not yet been fully established. Thus, a focus should be put on preventing PPPD and MdDS, especially in patients with personality traits or psychosocial states that places them at risk. In future prospective studies, the spectrum of the clinical phenotype of patients with prolonged vestibular symptoms and migrainous headaches needs to be further characterized, and different approaches to classify such patients should be compared. Importantly, this should include a screening for accompanying anxiety and panic disorders to characterize psychiatric comorbidity as well. A substantial discrepancy in the clinical presentation of this subgroup and patients with established PPPD diagnosis may indeed prompt establishing separate diagnostic criteria for a chronic variant of VM. This is also in line with the recent update of the classification committee of the Bárány Society in 2021, concluding that chronic vestibular migraine may become a formally recognized category of a revised classification [8].

## 8. Conclusions

Vestibular migraine, PPPD and MdDS lie on an episodic-to-chronic vestibular spectrum and may co-exist in a single patient. Specifically, when vestibular symptoms in patients with migraine headaches become chronic, they should—whenever possible—be diagnosed as PPPD. Moreover, attempts to identify those patients with subacute vestibular symptoms who are at increased risk for developing PPPD or MdDS should be intensified, and prospective treatment studies should be initiated. Ideally, the transition from episodic to chronic vestibular symptoms can be interrupted by targeted personalized interventions.

## Figures and Tables

**Figure 1 brainsci-13-01692-f001:**
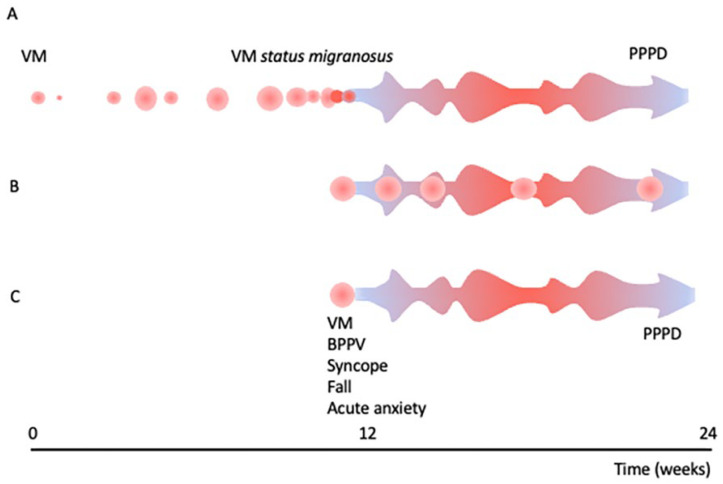
Vestibular migraine—episodic to chronic symptoms. This is a diagrammatic representation of the interplay and transition from vestibular migraine (VM), status migrainosus and persistent postural-perceptual dizziness (PPPD), proposing different paths for how chronic vestibular symptoms may arise. (**A**) The development of persistent postural-perceptual dizziness from recurrent episodes of vestibular migraine. (**B**) In patients with established PPPD, recurrent episodes of VM may occur on the background of chronic vestibular symptoms. (**C**) Alternatively, PPPD can arise from a single episode that impairs or threatens normal balance (e.g., VM, BPPV, or syncope). The size of the circles represents the severity of the attack in time, and the colour, its severity in functional burden (red representing maximal symptom burden). Abbreviations: BPPV, benign paroxysmal positional vertigo; PPPD, persistent postural perceptual dizziness; VM, vestibular migraine.

**Table 1 brainsci-13-01692-t001:** Diagnostic criteria for vestibular migraine from the International Classification of Vestibular Disorders—all four criteria (A–D) must be fulfilled to make the diagnosis (modified after [8,9]).

A. At least 5 episodes with vestibular symptoms of moderate or severe intensity, lasting 5 min to 72 h
B. Current or previous history of migraine with or without aura according to the International Classification of Headache Disorders (ICHD-3)
C. One or more migraine features with at least 50% of the vestibular episodes: – headache with at least two of the following characteristics: one sided location, pulsating quality, moderate or severe pain intensity, aggravation by routine physical activity – photophobia and phonophobia – visual aura
D. Not better accounted for by another vestibular or ICHD diagnosis

**Table 2 brainsci-13-01692-t002:** Diagnostic criteria for persistent postural-perceptual dizziness from the International Classification of Vestibular Disorders—all five criteria (A–E) must be fulfilled to make the diagnosis (modified after [16]).

A. One or more symptoms of dizziness, unsteadiness, or non-spinning vertigo are present on most days for 3 months or more Symptoms last for prolonged (hours-long) periods of time but may wax and wane in severity Symptoms need not be present continuously throughout the entire day
B. Persistent symptoms occur without specific provocation, but are exacerbated by three factors: Upright posture Active or passive motion without regard to direction or position Exposure to moving visual stimuli or complex visual patterns
C. The disorder is precipitated by conditions that cause vertigo, unsteadiness, dizziness, or problems with balance including acute, episodic, or chronic vestibular syndromes; other neurologic or medical illnesses; or psychological distress: When the precipitant is an acute or episodic condition, symptoms settle into the pattern of criterion A as the precipitant resolves, but they may occur intermittently at first, and then consolidate into a persistent course When the precipitant is a chronic syndrome, symptoms may develop slowly at first and worsen gradually
D. Symptoms cause significant distress or functional impairment
E. Symptoms are not better accounted for by another disease or disorder

**Table 3 brainsci-13-01692-t003:** Diagnostic criteria for mal de débarquement syndrome from the International Classification of Vestibular Disorders—all five criteria (A–E) must be fulfilled to make the diagnosis (modified after [17]).

A. Non-spinning vertigo characterized by an oscillatory perception (‘rocking’, ‘bobbing’, or ‘swaying’) present continuously or for most of the day
B. Onset occurs within 48 h after the end of exposure to passive motion
C. Symptoms temporarily reduce with exposure to passive motion
D. Symptoms continue for >48 h D0: MdDS in evolution: symptoms are ongoing but the observation period has been less than 1 month D1: Transient MdDS: symptoms resolve at or before 1 month and the observation period extends at least to the resolution point D2: Persistent MdDS: symptoms last for more than 1 month
E. Symptoms not better accounted for by another disease or disorder.

## Data Availability

This manuscript does not contain any new data.

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
