# Peer review of "What’s in a Name? Chronic Vestibular Migraine or Persistent Postural Perceptual Dizziness?"

_brainsci, 2023, doi:10.3390/brainsci13121692_

Round 1

Reviewer 1 Report

Comments and Suggestions for Authors

This is an interesting and thought provoking narrative review of the potential diagnostic criteria for a chronic variant of vestibular migraine. 

The process for identifying the literature search could be expanded, such as years considered, language, study design etc and how it was conducted (e.g. by bother authors etc). Only two of your search terms considered migraine ('vestibular migraine' and 'migraine') but many different terms have been used to describe vestibular migraine, so how can you be sure you did not miss anything - you say search terms 'including' making it sound like the search strategy was very broad/non-systematic and this should be addressed in the discussion as a limitation of this review. The paper would benefit from including a limitations section.  

The authors argue that current descriptions of chronic vestibular migraine (CVM) are insufficient and instead that most of these patients belong to PPPD. However, when reviewing such a limited field of research the authors risk hasty generalizations at several points in the paper. E.g. line 145-148 they generalize the findings of Oh (2015) to make the point they want to make. When confronted with data that does not align with their argument, specifically that patients with VM did not meet the criteria for PPPD regarding upright posture, they propose changing the PPPD criteria - but does this not then weaken diagnosis of PPPD and risk the perception of a 'waste-basket syndrome'? 

Likewise, there is an 'either or' fallacy in the argument that by recognizing a CVM label clinicians are not optimized to recognize patients at risk (line 337) or offer a holistic approach (line 301). One argument would be that recognizing CVM does indeed optimize clinicians to recognize this transition earlier, if attacks or symptoms are lasting longer than is expected of VM. And management of chronic migraine is often holistic, including appropriate preventative medications as well as CBT and lifestyle changes, but labelling it as PPPD might prevent the migraine from being addressed. 

The authors argue on lines 296-299 that a distinction between CVM and PPPD seems to only be a semantic deliberation that evades the clinical focus. However, isn't the point of labelling correctly to steer treatment and response to treatment? The argument ignores the potential migraine related mechanisms that might lead to persistent symptoms such as lack of habituation. 

Some references could be improved - for example there must be a better reference for psych/physical therapy than Popkirov 'in press# (line 252). They also do not address prior arguments of a 'Migraine-Anxiety Related Dizziness' (see Furman et al 2004). 

The authors also do not address the argument of CVM as outlined in the latest Barany document - that with more research CVM could be recognized. Since the literature is so limited in this area, a more constructive and expanded 'future directions' sections could outline what gaps in the literature they have found in the review, and specifically what the authors believe future research should be done and what that research would need to address to prove a difference between CVM and PPPD? I say this, because clinically I do come across patients with chronic symptoms in the context of vestibular migraine (specifically chronic self motion intolerances, but often normal balance, with episodic fluctuations) that are distinct from the PPPD cohort (who typically are more anxious, higher threat postural controls etc) - for a whom a different vestibular therapy protocol may be needed, hence I do not quite buy the argument that any differences are not clinically meaningful. 

Author Response

Reviewer 1

This is an interesting and thought-provoking narrative review of the potential diagnostic criteria for a chronic variant of vestibular migraine. 

The process for identifying the literature search could be expanded, such as years considered, language, study design etc and how it was conducted (e.g. by bother authors etc). Only two of your search terms considered migraine ('vestibular migraine' and 'migraine') but many different terms have been used to describe vestibular migraine, so how can you be sure you did not miss anything - you say search terms 'including' making it sound like the search strategy was very broad/non-systematic and this should be addressed in the discussion as a limitation of this review. The paper would benefit from including a limitations section.  

Reply by the authors: We thank the reviewer for this important remark. Indeed, we did not perform a systematic review, thus we cannot confidently exclude having missed relevant publications. We did however screen several other topical reviews on migrainous vertigo, migraine associated dizziness, vestibular migraine, PPPD and mal de débarquement syndrome for additional publications, and have included this in the search strategy section:

" We have also screened published consensus papers and reviews on vestibular migraine (and related terms), PPPD and MdDS for further relevant citations.”

We have added a limitations section as suggested at the end of the article:

“As this was a critical review, we did not perform a systematic review of the literature. Therefore, we cannot exclude that we may have missed relevant publications. Recognising this risk, we screened published review articles on vestibular migraine (and its predecessor terms), PPPD and MdDS.”

The authors argue that current descriptions of chronic vestibular migraine (CVM) are insufficient and instead that most of these patients belong to PPPD. However, when reviewing such a limited field of research the authors risk hasty generalizations at several points in the paper. E.g. line 145-148 they generalize the findings of Oh (2015) to make the point they want to make. When confronted with data that does not align with their argument, specifically that patients with VM did not meet the criteria for PPPD regarding upright posture, they propose changing the PPPD criteria - but does this not then weaken diagnosis of PPPD and risk the perception of a 'waste-basket syndrome'? 

Reply by the authors: We thank the reviewer for this important and thought-provoking remark. We have considered this comment carefully. We have identified only two publications that proposed (distinct) criteria for a chronic variant of VM that is, of course, not included in the classification of the international headache society (IHS). We wanted to make sure we addressed this proposed new category, and our intention was to provide a balanced view. We are grateful therefore for your comment. We have now revised several sections of the manuscript to provide a more balanced view. For example, in relation to the findings by Oh et al. (2015), we have softened this statement to read: “Thus, this subset of patients with chronic dizziness presented with recurrent exacerbations of their vertigo or dizziness in conjunction with migraine headaches, that closely resembles PPPD with superimposed VM rather than a single distinct entity of chronic dizziness.”

In reference to the Chae et al. manuscript on page 5 we have revised the sentence thus:

“Noteworthy, diagnostic criteria for PPPD were not met in the vast majority of patients studied by Chae and colleagues because of absent worsening of symptoms with upright posture, suggesting there may be subtle but distinct pathophysiological mechanisms across CVM and PPPD or indeed that this feature of PPPD may be too restrictive.”

Furthermore, suggesting a change to the PPPD criteria has been reframed to try to reduce the bias:

“One potential strategy would be to revise current PPPD criteria, critically reviewing whether the required worsening of symptoms when standing up is a strict defining feature of PPPD or whether it could be considered optional. Alternatively, future clinical data may identify distinct pathophysiological mechanisms in patients with VM who develop persistent dizziness, versus those that develop PPPD in the absence of migraine features.”

Finally, we have added the following sentence to the limitation section recognizing the need for further work to best define diagnostic criteria in chronic dizziness:

“Importantly, the existing literature on the transition from episodic vestibular syndromes to chronic vestibular syndromes is limited and existing diagnostic criteria (for VM and PPPD) may have neglected certain subgroups including those with prolonged / persistent vestibular symptoms not fitting the current PPPD criteria. Thus, future, prospective studies are needed to gain more knowledge about the clinical presentation and the disease course of such subgroups.”

Likewise, there is an 'either or' fallacy in the argument that by recognizing a CVM label clinicians are not optimized to recognize patients at risk (line 337) or offer a holistic approach (line 301). One argument would be that recognizing CVM does indeed optimize clinicians to recognize this transition earlier, if attacks or symptoms are lasting longer than is expected of VM. And management of chronic migraine is often holistic, including appropriate preventative medications as well as CBT and lifestyle changes, but labelling it as PPPD might prevent the migraine from being addressed. 

Reply by the authors: Again, an excellent point. We have reflected on this and agree with the reviewer that the diagnostic label of CVM does not preclude a holistic approach to treatment. Our argument is that the inclusion of the term ‘migraine’ (whether chronic or not) may be misconstrued to imply that treatment is the same as for an episodic migraine disorder. This has certainly been our independent experience across the two centres. We do think that increasing the emphasis on early recognition of patients at risk for developing chronic vestibular symptoms is important. We have emphasized in the revised manuscript that proposed diagnostic criteria for a chronic variant of VM require a minimal symptom duration of 3-6 months and that it may be important to initiate a holistic treatment approach before this time point. In this regard, introducing the concept of “in transition to a chronic vestibular syndrome” as for those established for MdDS, could be of help. The relevant section now reads:

“With regards to the diagnostic criteria proposed for a chronic variant of VM, they are not optimized to recognize patients at risk. Rather they will detect patients that have already advanced to a (sub)chronic stage, with a symptom duration of at least 3-6 months. Consideration of the concept of “in transition to a chronic vestibular syndrome” as for those established for MdDS, could help shift the focus to prevention of chronification.”

The authors argue on lines 296-299 that a distinction between CVM and PPPD seems to only be a semantic deliberation that evades the clinical focus. However, isn't the point of labelling correctly to steer treatment and response to treatment? 

Reply by the authors: Admittedly, there are different strategies to classify patients with chronic vestibular symptoms and migrainous headaches. One could provide separate diagnostic criteria for PPPD and for a chronic variant of VM (as proposed by Chae and colleagues in their 2021 publication) or one could to modify existing diagnostic criteria covering chronic vestibular syndromes (i.e., PPPD). While providing separate diagnostic criteria for a chronic vestibular migraine variant and for PPPD bears the risk of substantial overlap, implementing less rigid diagnostic criteria for PPPD may result in a blurring of the patient cohorts included. Based on our critical literature review we were in favor of the second approach because both CVM and PPPD would share the same treatment framework. We do agree however that there is insufficient data to draw firm conclusions at this stage and have therefore softened our recommendations regarding future study directions:

“In future prospective studies the spectrum of the clinical phenotype of patients with prolonged vestibular symptoms and migrainous headaches needs to be further characterized and different approaches to classify such patients should be compared. Importantly, this should include a screening for accompanying anxiety and panic disorders to characterize the psychiatric comorbidity as well. A substantial discrepancy in the clinical presentation of this subgroup and patients with established PPPD diagnosis may indeed prompt establishing separate diagnostic criteria for a chronic variant of VM.”

The argument ignores the potential migraine related mechanisms that might lead to persistent symptoms such as lack of habituation.

The Reviewer raises an interesting point here. We do not think that there exists sufficient evidence behind the differences (if any) of the mechanisms that drive persistent dizziness symptoms in VM versus other triggers (BPPV, poorly compensated unilateral peripheral vestibulopathy etc.). We propose that there may, again, be overlapping mechanisms between those that develop PPPD from a peripheral vestibular trigger and those that arrive at this juncture from recurrent VM episodes. We have made reference to this in the discussion:

“Abnormal processing of repeated external stimuli (e.g., visual information) in the inter-ictal phase is a recognized pathophysiological feature of migraine, known as habituation deficiency [39]. Such features may play in role in the chronification of dizziness symptoms in VM.”

Some references could be improved - for example there must be a better reference for psych/physical therapy than Popkirov 'in press# (line 252). They also do not address prior arguments of a 'Migraine-Anxiety Related Dizziness' (see Furman et al 2004). 

Reply by the authors: We have added a citation from a review article from Jeffrey Staab (Semin Neurol 2020) that summarized current treatment approaches in PPPD, emphasizing the value of cognitive behavioral therapy (CBT) early on. We would like to keep the paper in press citation (Popkirov et al. 2023) also as this should be fully published soon and has reviewed the relevant literature so it serves as a helpful signpost to interested readers. We have rephrased the sentence under discussion accordingly and have added another citation (Herdman et al. 2022):

“Unlike treatment for VM where prophylactic agents are used to reduce the frequency and severity of attacks, therapeutic strategies for PPPD work to reduce bodily hypervigilance and normalize mis-processing of self and visual motion signals through gradual re-exposure, thus emphasizing the value of cognitive physical therapy approaches early on [26] or even their preferential use over medications (see [34] and [Popkirov et al. in press]).”

With regards to the 2005 publication from Furman and colleagues on MARD we do agree that there is a significant link between migraine, anxiety disorders and dizziness. This is also reflected in treatment approaches established both in PPPD and (vestibular) migraine, including antidepressants, antiepileptics (with mood stabilizing effect such as lamotrigine) and CBT. We have thus added the following sentence at the beginning of the discussion (end of first paragraph), citing the work from Furman and colleagues (2005):

“The important interplay between migraine, anxiety and dizziness has been elegantly described in the literature [35] and such features may be help guide individualised therapies.”

The authors also do not address the argument of CVM as outlined in the latest Barany document - that with more research CVM could be recognized. Since the literature is so limited in this area, a more constructive and expanded 'future directions' sections could outline what gaps in the literature they have found in the review, and specifically what the authors believe future research should be done and what that research would need to address to prove a difference between CVM and PPPD? I say this, because clinically I do come across patients with chronic symptoms in the context of vestibular migraine (specifically chronic self motion intolerances, but often normal balance, with episodic fluctuations) that are distinct from the PPPD cohort (who typically are more anxious, higher threat postural controls etc) - for a whom a different vestibular therapy protocol may be needed, hence I do not quite buy the argument that any differences are not clinically meaningful. 

Reply by the authors: We thank the reviewer for this important remark. We definitely do agree that the literature on the spectrum of clinical phenotypes for patients with chronic migraine and dizziness, PPPD and migraine or PPPD alone is scarce. This is a clear limitation and needs to be addressed in the future.

We have addressed the Discussion paragraph that now reads:

“A central theme that requires exploration is whether the labelling persistent vestibular symptoms as chronic VM or PPPD is a semantic deliberation with little clinical impact, or whether there are indeed fundamental clinical differences (greater anxiety, higher postural threat etc.) requiring different treatment protocols. Ultimately, however, it is the underlying mechanism of the symptoms that will dictate the most appropriate therapeutic approach to maximise chances of a positive outcome. Treating a patient with persistent migrainous symptoms with preventative medications (for VM), is less likely to be effective than a more holistic approach that targets the persistent perceptual and psychological variables, whatever the label chosen. The importance of the interplay between migraine, anxiety and dizziness has been emphasized in the literature as well [35].”

We have added greater emphasis on this in the future directions section:

“Furthermore, in future prospective studies the spectrum of the clinical phenotype of patients with prolonged vestibular symptoms and migrainous headaches needs to be further characterized and different approaches to classify such patients should be compared. Importantly, this should include a screening for accompanying anxiety and panic disorders to characterize the psychiatric comorbidity as well. A substantial discrepancy in the clinical presentation of this subgroup and patients with established PPPD diagnosis may indeed prompt establishing separate diagnostic criteria for a chronic variant of VM. This is also in line with the recent update of the classification committee of the Bárány Society in 2021, concluding that chronic vestibular migraine may become a formally recognized category of a revised classification [8].”

Reviewer 2 Report

Comments and Suggestions for Authors

I appreciate the chance to conduct a peer review of the article titled "What’s in a name? Chronic vestibular migraine or persistent postural perceptual dizziness?". This manuscript was also previously posted on Preprints.org several days earlier. The manuscript is centered around utilizing current evidence to address the issue (chronic variant of vestibular migraine - VM) outlined in the Diagnosis criteria for vestibular migraine, as presented in the initial publication in 2012 and the revised version in 2022 by the Consensus document of the Bárány Society and the International Headache Society. Specifically, the authors summarized current evidence of such a potential chronic variant of VM and critically reviewed proposed definitions for chronic variant of VM (CMV). The authors suggested potential approaches to establish a diagnostic framework for CVM including (a) following the distinction between episodic and chronic migraine headaches; namely, frequent and/or prolonged episodes of VM in the context of chronic migraine headaches or (b) daily dizzy spells over more than 6 months that responded well to prophylactic anti-migraine therapy. They recommended intensifying efforts to identify patients with subacute vestibular symptoms at an elevated risk of developing persistent postural-perceptual dizziness (PPPD) or mal de débarquement syndrome (MdDS), emphasizing the initiation of prospective treatment studies to interrupt the transition from episodic to chronic vestibular symptoms through targeted personalized interventions.

The manuscript is interesting and well written. It instills hope for arriving at a suitable diagnosis in cases where a patient displays features of VM but vestibular symptoms become chronic (lasting more than 72 hours), a scenario occasionally encountered in clinical practice. I offer only a few minor recommendations for consideration:

1. Since the diagnostic criteria of VM (criteria A to C) rely on patient self-report, distinguishing between consecutive episodic (vestibular symptom) attacks and prolonged attacks can be challenging. Could the authors please give some discussion on how to solve this problem?

2. It appears that the manuscript intends to highlight the clinical course of episodic flares of vestibular symptoms in VM/CVM superimposed on a background of persistent dizziness. However, PPPD/MdDS is not a diagnosis of exclusion; they have strict diagnostic criteria (even without a probable type). Therefore, the question arises as to why the authors aim to link the prolonged vestibular symptoms in CVM to PPPD/MdDS rather than other forms of persistent/chronic dizziness? Is the manuscript title "Chronic Vestibular Migraine or Persistent Postural Cognitive Vertigo?" appropriate?

3. Vestibular migraine is recognized as both a precipitant and a co-morbidity of PPPD, as indicated by studies such as Staab (2020,"Persistent postural-perceptual dizziness." Semin Neurol) and Waterston et al. (2021, "Persistent postural-perceptual dizziness: precipitating conditions, co-morbidities and treatment with cognitive behavioral therapy." Front Neurol). What is the distinction between a patient developing PPPD with VM/CVM and patients developing PPPD without VM/CVM among those experiencing dizziness?

4. "Taking together the clinical data available, emphasizing the substantial overlap between chronic functional vestibular disorders... By so doing, patients with persistent or very frequent vestibular symptoms and migraine headaches could be diagnosed under the PPPD umbrella." It would be nice if this paragraph offered a more comprehensive overview, emphasizing that extended vestibular symptoms within the context of VM/CVM are associated with persistent/chronic dizziness in a broader sense, rather than the scope of only PPPD.

5. Please give a title for Figure 1. The author wrote: "The size of the circles represents the severity of the attack in time, and the colour, its severity in functional burden (red representing maximal symptom burden)". Please provide references/resources for this statement.

Minor comments:

- Because there is a consensus document on "mal de débarquement syndrome" [Cha et al. (2020). Mal de débarquement syndrome diagnostic criteria: Consensus document of the Classification Committee of the Barany Society. J Vestib Res], so it is better to replace "mal-de-débarquement syndrome" with the formal term "mal de débarquement syndrome" (line 21, 22)

- The phrase "including migraine headaches in at least every second attack" (line 40) appears to convey a distinct meaning compared to the diagnostic criteria "One or more migraine features with at least 50% of the vestibular episodes."

- Please give the full term of “HIS” (line 83), “EVS” “CVS” (line 333)

- Please capitalize "vm" as VM (line 149)  

- Line 334: phrase “the six month minimal symptom duration for PPPD is reached” is incorrect.

Author Response

Reviewer 2

I appreciate the chance to conduct a peer review of the article titled "What’s in a name? Chronic vestibular migraine or persistent postural perceptual dizziness?". This manuscript was also previously posted on Preprints.org several days earlier. The manuscript is centered around utilizing current evidence to address the issue (chronic variant of vestibular migraine - VM) outlined in the Diagnosis criteria for vestibular migraine, as presented in the initial publication in 2012 and the revised version in 2022 by the Consensus document of the Bárány Society and the International Headache Society. Specifically, the authors summarized current evidence of such a potential chronic variant of VM and critically reviewed proposed definitions for chronic variant of VM (CMV). The authors suggested potential approaches to establish a diagnostic framework for CVM including (a) following the distinction between episodic and chronic migraine headaches; namely, frequent and/or prolonged episodes of VM in the context of chronic migraine headaches or (b) daily dizzy spells over more than 6 months that responded well to prophylactic anti-migraine therapy. They recommended intensifying efforts to identify patients with subacute vestibular symptoms at an elevated risk of developing persistent postural-perceptual dizziness (PPPD) or mal de débarquement syndrome (MdDS), emphasizing the initiation of prospective treatment studies to interrupt the transition from episodic to chronic vestibular symptoms through targeted personalized interventions.

The manuscript is interesting and well written. It instills hope for arriving at a suitable diagnosis in cases where a patient displays features of VM but vestibular symptoms become chronic (lasting more than 72 hours), a scenario occasionally encountered in clinical practice. I offer only a few minor recommendations for consideration:

  1. Since the diagnostic criteria of VM (criteria A to C) rely on patient self-report, distinguishing between consecutive episodic (vestibular symptom) attacks and prolonged attacks can be challenging. Could the authors please give some discussion on how to solve this problem?

Reply by the authors: We do agree with the reviewer that receiving a detailed description of the duration and frequency of vestibular symptoms can be challenging. Thus, history taking needs to be performed in a very structured way, asking how long single episodes with vestibular symptoms lasted and how frequent such episodes occurred. In this context, it is important to distinguish between conditions with continuous (baseline) vestibular symptoms and episodic exacerbations (as one would typically expect in PPPD patients) and episodic vestibular symptoms with symptom-free intervals (as in VM). We have added a statement in the limitations section emphasizing this:

“For patients distinguishing between conditions with episodic vestibular symptoms (and symptom-free intervals) and chronic vestibular symptoms with varying intensity may be challenging. Thus, structured history taking addressing specifically the duration of single episodes with vestibular symptoms, their frequency and the presence / absence of symptom-free intervals is important to decide whether the diagnostic criteria for VM or PPPD are met. This will also be critical for future studies further delineating the clinical spectrum of patients with prolonged vestibular symptoms and VM.”   

  1. It appears that the manuscript intends to highlight the clinical course of episodic flares of vestibular symptoms in VM/CVM superimposed on a background of persistent dizziness. However, PPPD/MdDS is not a diagnosis of exclusion; they have strict diagnostic criteria (even without a probable type). Therefore, the question arises as to why the authors aim to link the prolonged vestibular symptoms in CVM to PPPD/MdDS rather than other forms of persistent/chronic dizziness? Is the manuscript title "Chronic Vestibular Migraine or Persistent Postural Cognitive Vertigo?" appropriate?

Reply by the authors: We are grateful to this comment that highlights a degree of lack of clarity on our part. The intention here is to avoid unnecessary additional diagnostic categories for what may be the same pathophysiological mechanism (persistent dizziness). We are in fact suggesting that CVM may not be a necessary entity at all, as the symptoms are mostly compatible with PPPD. We have attempted to address this throughout the manuscript. For example:

“A central theme that requires exploration is whether the labelling persistent vestibular symptoms as chronic VM or PPPD is a semantic deliberation with little clinical impact, or whether there are indeed fundamental clinical differences (greater anxiety, higher postural threat etc) requiring different treatment protocols. Ultimately however, it is the underlying mechanism of the symptoms that will dictate the most appropriate therapeutic approach to maximise chances of a positive outcome.”

  1. Vestibular migraine is recognized as both a precipitant and a co-morbidity of PPPD, as indicated by studies such as Staab (2020,"Persistent postural-perceptual dizziness." Semin Neurol) and Waterston et al. (2021, "Persistent postural-perceptual dizziness: precipitating conditions, co-morbidities and treatment with cognitive behavioral therapy." Front Neurol). What is the distinction between a patient developing PPPD with VM/CVM and patients developing PPPD without VM/CVM among those experiencing dizziness?

Reply by the authors: Thank you for the comment. This is a question also discussed in Reviewer 1 comments. There is currently insufficient evidence to help us understand whether there are differences between arriving at PPPD through VM or via another trigger. Reviewer 1 correctly points out that there may be subtle clinical differences and that these may be relevant for treatment. We have added the following statement in the ‘future directions’ section:

“Furthermore, in future prospective studies the spectrum of the clinical phenotype of patients with prolonged vestibular symptoms and migrainous headaches needs to be further characterized and different approaches to classify such patients should be compared. Importantly, this should include a screening for accompanying anxiety and panic disorders to characterize the psychiatric comorbidity as well. A substantial discrepancy in the clinical presentation of this subgroup and patients with established PPPD diagnosis may indeed prompt establishing separate diagnostic criteria for a chronic variant of VM. This is also in line with the recent update of the classification committee of the Bárány Society in 2021, concluding that chronic vestibular migraine may become a formally recognized category of a revised classification [8].”

  1. "Taking together the clinical data available, emphasizing the substantial overlap between chronic functional vestibular disorders... By so doing, patients with persistent or very frequent vestibular symptoms and migraine headaches could be diagnosed under the PPPD umbrella." It would be nice if this paragraph offered a more comprehensive overview, emphasizing that extended vestibular symptoms within the context of VM/CVM are associated with persistent/chronic dizziness in a broader sense, rather than the scope of only PPPD.

Reply by the authors: Thank you. This is again an interesting point. We have considered this carefully. We have decided to stick to the established diagnostic criteria for persistent dizziness so as not to muddy the waters. We think this links to the point above, and hopefully further research in this area will help clarify whether extended vestibular symptoms in VM are indeed encapsulated by PPPD or differ to this in some ways.

  1. Please give a title for Figure 1. The author wrote: "The size of the circles represents the severity of the attack in time, and the colour, its severity in functional burden (red representing maximal symptom burden)". Please provide references/resources for this statement.

 Reply by the authors: The Figure was intended as a diagrammatic representation of common clinical presentations. There are therefore no relevant citations for this. We have explained this in the revised figure legend.

We propose the following title for figure 1:

“Vestibular migraine – episodic to chronic symptoms. This is a diagrammatic representation of the interplay and transition from vestibular migraine (VM), status migrainosus and persistent postural-perceptual dizziness (PPPD), proposing different paths to how chronic vestibular symptoms may arise.”

Minor comments:

- Because there is a consensus document on "mal de débarquement syndrome" [Cha et al. (2020). Mal de débarquement syndrome diagnostic criteria: Consensus document of the Classification Committee of the Barany Society. J Vestib Res], so it is better to replace "mal-de-débarquement syndrome" with the formal term "mal de débarquement syndrome" (line 21, 22)

Reply by the authors: Thank you. Done

- The phrase "including migraine headaches in at least every second attack" (line 40) appears to convey a distinct meaning compared to the diagnostic criteria "One or more migraine features with at least 50% of the vestibular episodes."

Reply by the authors: done. The sentence now reads: “Importantly, VM was conceptualized as an episodic disorder with recurrent dizzy spells lasting between 5 minutes and 72 hours and accompanying migrainous features with at least 50% of the vestibular episodes (Box 1).”

- Please give the full term of “HIS” (line 83), “EVS” “CVS” (line 333)

Reply by the authors: This has been amended, thank you.

- Please capitalize "vm" as VM (line 149)  

Reply by the authors: Done, thank you.

- Line 334: phrase “the six month minimal symptom duration for PPPD is reached” is incorrect.

Reply by the authors: done. Thanks for pointing this out!

Reviewer 3 Report

Comments and Suggestions for Authors

Thank you for submitting your interesting review paper.

Only a view comments:

L83: what does HIS stand for?

L119: what does Md stand for? (I guess its MdDS. In this case please use the same abbreviation in the text...)

Figure 1: please use (A) , (B) and (C) before the sentence and not after. Its confusing.

And in general I would prefer a view more words about treatment of these conditions and if you could compare the treatment options (for VM vs. PPPD vs. MdDS)

Author Response

Reviewer 3

Thank you for submitting your interesting review paper.

Only a view comments:

L83: what does HIS stand for?

Reply by the authors: this is a typo (due to word auto-correction). It should be IHS for International Headache Society. We have changed this.

L119: what does Md stand for? (I guess its MdDS. In this case please use the same abbreviation in the text...)

Reply by the authors: MD stands for Menière’s disease. The abbreviation is provided when first mentioned:

“They focused on the spectrum of clinical vestibular presentations and diagnoses (such as PPPD, MdDS, benign paroxysmal positional vertigo [BPPV] or Menière’s disease [MD]) but did not state whether they recruited consecutive patients, thus were potentially open to enrichment of chronic VM patients.”

Figure 1: please use (A) , (B) and (C) before the sentence and not after. Its confusing.

Reply by the authors: we agree and have modified this accordingly.

And in general I would prefer a view more words about treatment of these conditions and if you could compare the treatment options (for VM vs. PPPD vs. MdDS)

Reply by the authors: We provide more citations to publications focusing on the spectrum of treatment options in these distinct conditions in the discussion section.

“Moreover, current treatment approaches for VM [24], PPPD [26] and MdDS [40] are overlapping in relation to pharmacotherapy, especially when considering antidepressant medications.”

However, we are conscious of the need to keep the focus of this paper in mind and do not think it would be appropriate to provide a narrative of treatment options for the various conditions. We hope that the addition of further references will act as a signpost to interested readers. 

Reviewer 4 Report

Comments and Suggestions for Authors

Thank you for inviting me to review this manuscript about a review on vestibular migraine. The concepts are clear and well illustrated with tables. I highlight the complete review of the previous and actual classifications and the details regarding clinical findings and associated differential diagnoses. I'd suggest the authors add a graphic illustration/cartoon to enhance this submission and make it more attractive for readers. It could add some anatomical/physiological aspects regarding vestibular migraine, PPPD, and MdDS. 

Author Response

Reviewer 4:

Thank you for inviting me to review this manuscript about a review on vestibular migraine. The concepts are clear and well-illustrated with tables. I highlight the complete review of the previous and actual classifications and the details regarding clinical findings and associated differential diagnoses. I'd suggest the authors add a graphic illustration/cartoon to enhance this submission and make it more attractive for readers. It could add some anatomical/physiological aspects regarding vestibular migraine, PPPD, and MdDS. 

Reply by the authors: We thank the reviewer for his/her positive feedback. We have evaluated the option of adding a graphical illustration / abstract as defined in the instruction for the authors based on the reviewer’s suggestion. While we see the potential value of such illustrations in original publications, we would prefer not to provide such an illustration for this critical review. Note that this Review already has a Figure that depicts the different stages of vestibular migraine, from episodic to chronic. With regards to adding more anatomical / physiological aspects regarding VM, PPPD and MdDS we wonder whether the Reviewer and Editorial team would be happy for us to provide suitable references to signpost interested readers, so as not to detract from the main focus of this review.